# Salvage Radical Prostatectomy after Primary Focal Ablative Therapy: A Systematic Review and Meta-Analysis

**DOI:** 10.3390/cancers15102727

**Published:** 2023-05-12

**Authors:** Fernando Blank, Meredith Meyer, Hannah Wang, Hasan Abbas, Shima Tayebi, Wei-Wen Hsu, Abhinav Sidana

**Affiliations:** 1Division of Urology, Department of Surgery, University of Cincinnati Medical Center, Cincinnati, OH 45267, USA; blankvfo@mail.uc.edu (F.B.);; 2University of Cincinnati College of Medicine, Cincinnati, OH 45267, USA; 3Division of Biostatistics and Bioinformatics, University of Cincinnati College of Medicine, Cincinnati, OH 45267, USA

**Keywords:** prostate cancer, focal therapy, salvage radical prostatectomy, recurrence

## Abstract

**Simple Summary:**

Focal therapy is a treatment modality option for select patients with localized intermediate-risk prostate cancer. A rise in its use over recent years has been brought on by its favorable side effect profile, mainly the reduced risk of erectile dysfunction or continence over the current standard of through either radiotherapy or radical prostatectomy. Mainly still in its early stages of use, a notable challenge with this treatment modality is the significant risk of local cancer recurrence requiring subsequent treatments. While patients have several options for further treatment, some will opt to undergo salvage radical prostatectomy. However, the data is lacking for postoperative, oncologic, and functional outcomes for these patients. Thus, we performed a systematic review on patients who underwent radical prostatectomy for prostate cancer recurrence after prior failed primary focal therapy. Our overall findings showed acceptable complication rates and oncologic outcomes, however, with suboptimal functional outcomes for patients undergoing sRP for recurrent PCa after prior FT. Inferior outcomes were also observed for salvage treatment compared to primary radical prostatectomy (pRP). These findings are critical, as it will ultimately determine treatment modality choice after FT failure for future PCa patients.

**Abstract:**

Context: Focal therapy (FT) has been gaining popularity as a treatment option for localized intermediate-risk prostate cancer (PCa) due to the associated lower morbidity compared to whole-gland treatment. However, there is an increased risk of local cancer recurrence requiring subsequent treatment in a small proportion of patients. Objective: To conduct a systematic review and meta-analysis to better describe and analyze patient postoperative, oncologic, and functional outcomes for those who underwent salvage radical prostatectomy (sRP) to manage their primary FT failure. Evidence acquisition: A systematic review was completed using three databases (PubMed, Embase, and CINAHL) from October to December 2021 to identify data on outcomes in patients who received sRP for cancer recurrence after prior focal treatment. Evidence synthesis: 12 articles (482 patients) were included. Median time to sRP was 24 months. Median follow-up time was 27 months. A meta-analysis revealed a postoperative complication rate of 15% (95% CI: 0.09, 0.24), with 4.6% meeting criteria for a major complication Clavien (CG) grade ≥3. Severe GU toxicity was seen in 3.6% of the patients, and no patients had severe GI toxicity. Positive surgical margins (PSM) were found in 27% (95% CI: 0.19, 0.37). Biochemical recurrence (BCR) after sRP occurred in 23% (95% CI: 0.17, 0.30), indicating a BCR-free probability of 77% at 2 years. Continence (pad-free) and potency (ability to have penetrative sex) were maintained in 67% (95% CI: 0.53, 0.78) and 37% (95% CI: 0.18, 0.62) at 12 months, respectively. Conclusion: Our evidence shows acceptable complication rates and oncologic outcomes; however, with suboptimal functional outcomes for patients undergoing sRP for recurrent PCa after prior FT. Inferior outcomes were observed for salvage treatment compared to primary radical prostatectomy (pRP). More high-quality studies are needed to better characterize outcomes after this sequence of PCa treatments. Patient summary: We looked at treatment outcomes and toxicity for men treated with sRP for prior FT failure. We conclude that these patients will have significant detriment to genitourinary function, with outcomes being worse than those for pRP patients.

## 1. Introduction

Focal therapy (FT) for localized intermediate-risk prostate cancer (PCa) is an increasingly utilized treatment option. It involves the use of various energy modalities to target and destroy affected prostatic tissue. Over recent years, FT has emerged as an alternative treatment option due to the advancements in imaging and tissue sampling techniques that have allowed for better spatialization of the cancer within the gland. The aim of the FT is to only target portions of the prostate gland with clinically significant cancer, thereby freeing up the patient from damage to surrounding neurovascular structures, which may ultimately lead to better functional outcomes. FT is often described as a middle-ground option between the current standard of care for PCa, which includes active surveillance (AS), and the radical whole-gland treatment options of radical prostatectomy (RP) and radiotherapy (RT), which have traditionally been the gold standard supported by research from the urologic bodies [1,2,3,4,5,6,7]. While FT has demonstrated excellent functional outcomes in men with PCa, data on long term oncologic outcomes is scarce [8]. Despite the favorable outlook on this treatment modality, there is a significant concern for higher cancer recurrence risk, as by preserving genitourinary function, much of the prostate gland is left intact [9]. For example, in a large multicenter study on patients receiving partial high-intensity focused ultrasound (HIFU), a reported 42% (20–60%) had a recurrence or failure after partial ablation [10]. This indicates that a significant proportion of patients will require salvage treatment.

There are no standardized guidelines for treating these cancer recurrences, and options for salvage management include repeat focal therapy, whole-gland ablative therapy, or using the standard approaches of sRP or sRT [9]. In the past, worse functional outcomes were described for men who had salvage prostatectomy after radiation failure [11]. It remains to be determined whether patients undergoing sRP after FT failure share similar outcomes. Thus, the goal of this systematic review and meta-analysis is to consolidate the data to help clarify the oncological benefit and potential toxicity of sRP after FT. The significance is that further understanding of if and how FT affects the efficacy of a subsequent prostatectomy within these realms may influence clinical decision making for post-FT PCa recurrences and might impact how urologists counsel potential FT patients. 

## 2. Evidence Acquisition

### 2.1. Search Strategy, Inclusion Criteria and Exclusion Criteria

We aimed to describe the perioperative, oncological, and functional outcomes for sRP on recurrent or persistent cancer following FT (sub-total, focal, hemi-gland, or partial ablation). A protocol was approved and registered on the Prospective Register of Systematic Reviews (PROSPERO) database (ID = CRD42021289078). A Preferred Reporting Items for Systematic Reviews and Meta-Analyses (PRISMA)-adhering systematic review was conducted. A web search was performed from October to December 2021 through PubMed, Embase, and the Cumulative Index to Nursing and Allied Health Literature (CINAHL) platforms. The following search terms were used: “prostatic neoplasms” AND (“ablation” OR “focal therapy” OR “cryosurgery” OR “cryotherapy” OR “focal cryotherapy” OR “high-intensity focused ultrasound ablation” OR “ultrasound, high-intensity focused, transrectal” OR “laser therapy” OR “electroporation” OR “photochemotherapy” OR “vascular-targeted photodynamic therapy” OR “laser interstitial therapy” OR “radiofrequency ablation” OR “brachytherapy” AND (“salvage prostatectomy” OR “robotic salvage prostatectomy” OR “salvage radical prostatectomy” OR “prostatectomy/adverse effects” OR “prostatectomy/methods” OR “salvage therapy”). Upon obtaining the initial search results, one author (M.M.) organized and de-duplicated all retrieved records to prepare for the screening process, and two authors (F.B. and H.W.) independently screened these records by reading the titles and abstracts to identify potential articles of interest. Studies were excluded if: (1) primary therapy was not specified as focal; (2) salvage therapy treatment was other than by RP; (3) primary outcomes of interest were not reported; (4) they were written in any other language besides English; (5) they were review articles or abstracts; (6) sample size was <10. The remaining potential articles were then further screened by skimming through each full-text, and those were also excluded. Discrepancies were solved by a third author (M.M.).

### 2.2. Data Extraction and Quality Assessment

Once all criteria were met and the final articles were identified, two authors (F.B. and H.W.) independently read each included article in whole and extracted data relevant to the primary outcomes being assessed. Extracted data included: study size, average age at time of sRP, time from treatment to sRP, average follow up, FT treatment modalities used, sRP technique used, nerve-sparing attempts, pre-sRP pathology, post-sRP specimen pathology, postoperative complications, severe GU and GI toxicities, positive surgical margins (PSM), biochemical recurrence rates (BCR), continence (pad-free) rates at 12 months, and potency rates at 12 months. Discrepancies were resolved by a third author (H.A). Study quality and risk of bias were independently assessed for all individual articles through 7 domains found in the Risk of Bias in Non-randomized Studies of Interventions ROBINS-I tool by two authors (F.B. and H.W.), with disagreements settled through consensus [12]. The categories for risk of bias judgments were low risk, moderate risk, serious risk, and critical risk of bias based on specific criteria. 

### 2.3. Outcomes Categorization 

Postoperative complications were graded via the Clavien–Dindo classification system. Biochemical recurrence after FT was defined using the American Society of Therapeutic Radiology and Oncology (ASTRO) Phoenix criteria, defined as a rise of PSA ≥2 ng/mL from nadir, and rising on 2 successive measurements [13]. Biochemical recurrence after sRP was defined using the American Urologic Society (AUA) guideline recommendations of a PSA rise of ≥0.2 ng/mL from nadir and rising on 2 successive measurements [14]. Continence was defined as being pad- and leak-free at 12 months. Erectile function (EF) was defined as preserved ability to maintain an erection sufficient for penetrative sex with or without the use of medical treatment. It is important to note that definitions varied between authors, with only a few using standardized questionnaires such as the IIEF-5/SHIM.

### 2.4. Data Analysis

Categorical variables are given using frequencies and percentages. To calculate all the outcomes, a decision had to be made with respect to the denominator considered. For example, some studies did not have complete outcome data for all their patients, so the denominator was adjusted to account for all available data. All complications were scored on a per-event basis rather than a per-patient basis. The denominator was adjusted for potency to account for only those that were preoperatively potent. All the outcomes were further analyzed by meta-analysis to incorporate heterogeneity across different studies. It is important to note that the analysis was performed at a study level rather than an individual patient level. The estimated proportions and the associated 95% confidence intervals were reported. To evaluate the heterogeneity in the meta-analysis, $I^{2}$ and Tau-square ($\tau^{2}$) statistics were reported. Specifically, these two statistics can estimate the extent of heterogeneity, for which a higher value of $\tau^{2}$ indicates high degrees of heterogeneity [15]. Similarly, a higher value of $\tau^{2}$, which is an estimate of the between-study variance in a random-effects meta-analysis, means higher degrees of heterogeneity. Typically, the value of $I^{2}$ greater than 75% or a $\chi^{2}$ test with a *p*-value <0.1 implies heterogeneity of treatment effects. Heterogeneity can be accommodated appropriately by the random effects model in the meta-analysis that can take into consideration the diverse situations of PCa patients with FT included in this review [16]. All meta-analyses were performed using R software (version 4.1.3) with the package of ‘meta’. Excel software was used to tabulate the data.

## 3. Evidence Synthesis

### 3.1. Literature Search Results

The flowchart according to PRISMA guidelines is summarized in Figure 1. The initial search identified 7569 total results. Of 112 candidate studies that remained after applying inclusion/exclusion criteria, subsequent full-text screening identified 12 full-text articles, published from 2015 to 2020, that reported outcomes on 482 patients who underwent sRP after prior failed FT [17,18,19,20,21,22,23,24,25,26,27,28]. In total, 100% of the data was available for oncologic and postoperative complication outcomes. Data for 467 (96.9%) and 455 (94.4%) patients were available for continence and potency outcomes, respectively.

### 3.2. General Features and Quality 

The overall quality of the studies was low to moderate, consisting only of prospective or retrospective cohort studies (Table 1). None were from randomized controlled trials (RCTs). Of the 12 studies selected, 9 were retrospective, and 3 were prospective. Of the retrospective studies, 5 were comparative. Studies [19,22,25,28] compared sRP-FT to sRP-RT, and [23] compared sRP-FT to pRP. Only [28] specifically describes RT as whole-gland, while it is uncertain whether patients in [19,22,25] received focal or whole-gland RT. One of the prospective studies was comparative [27], and compared sRP-FT to pRP. Testing for heterogeneity with $I^{2}$ and Tau-square *(*$\tau^{2}$*)* statistics yielded *p* < 0.01 in all cases, and the percentage of total variance attributable to heterogeneity of outcomes between studies, as measured with the $I^{2}$ and $\tau^{2}$ statistic, varied from 59% to 84%, and 0.1921–0.8189, respectively, for all reported outcomes. 

### 3.3. Primary Focal Therapy Characteristics

The study design and data extracted from each study are summarized in Table 2. A total of 482 patients received primary focal treatment with subsequent sRP across the series. Types of primary treatment received included HIFU (*n* = 295 (61%)), cryotherapy (*n* = 79 (16%)), vascular photodynamic therapy (*n* = 69 (14%)), irreversible electroporation (*n* = 17 (4%)), laser (*n* = 15 (3%)), PRX302 (topsalysin) (*n* = 4 (1%)), and brachytherapy (*n* = 3 (1%)). Average age at time of sRP was 64 (range 61–67) years. sRP was performed at a median time of 24 (range 11–61) months after FT. SRP was performed robotically in 399/468 (85%), open in 60/468 (13%), and laparoscopically in 9/468 (2%). Nerve-sparing was feasible in 260/468 (56%) patients, consisting of 145 (31%) unilateral and 115 (25%) bilateral. A summary of these results is found in Table 3.

### 3.4. Postoperative Outcomes

Intraoperative complications were rare, and no rectal injuries occurred. All studies reported on postoperative complications, which ranged from 4 to 57%. Among 482 patients, 59 (12.2%) experienced some sort of postoperative complication, of which 18 (4.6%) met the criteria for a major complication (CG ≥ 3), ranging from 0 to 31% across all studies (Table 3). Urinary tract infections, anastomotic leaks, and wound infections were the most commonly reported complications. Severe GU toxicity was reported in 3.6% of the patients, while 0% were reported for severe GI toxicity. A meta-analysis (Figure 2A) was subsequently performed on the total complication rate, which was found to be 15% (95% CI: 0.09, 0.24). 

### 3.5. Oncologic Outcomes

All studies reported on oncologic outcomes, with data available for all 482 patients (Table 3). The median follow-up time was 27 months (range 10–62). BCR and positive surgical margins (PSM) ranged from 0 to 41% and 5 to 47%, respectively. In the pooled analysis, BCR occurred in 118 (24.5%) patients, with final pathology after sRP showing a PSM in 126 (26.1%). A meta-analysis revealed an overall estimate of 23% (95% CI: 0.17, 0.30) for BCR (Figure 2B) and 27% (95% CI: 0.19, 0.37) for PSM (Figure 2C). The overall BCR-free probability was 77% at a median follow-up time of 27 months. Only one (0.21%) cancer-related death was reported, which occurred in [22], and five (1%) metastases in [20,28].

### 3.6. Functional Outcomes

Data was available for continence (*n* = 467) and potency (*n* = 455) across all 12 studies (Table 3). For potency, 351 patients were preoperatively potent. Continence rates ranged from 23 to 92%, and potency rates varied from 0 to 91%. None of the studies reported on preoperative urinary continence status of patients. Overall, pad-free postoperative urinary continence was achieved in 319 (68.3%) patients. A total of 107 (30.5%) preoperatively potent patients continued to have potency at 12 months, either spontaneously or with medical assistance. Two studies [21,23] had 0% potency rates. Meta-analysis revealed an overall continence (Figure 2D) rate estimate of 67% (95% CI: 0.53, 0.78) and potency (Figure 2E) rate estimate of 37% (95% CI: 0.18, 0.62).

### 3.7. Discussion and Limitations

PCa continues to be the number one cancer and the second leading cause of cancer mortality in men [29]. FT has emerged as a potential alternative treatment, thought to have improved urinary and sexual function outcomes over historically favored RP and RT treatments [30,31]. However, FT has been limited by a significant risk of cancer recurrence that will require patients to undergo further definitive intervention, often with sRP. Patients undergoing FT are likely to have PCa recurrence, indicating FT failure, however, the definition is not standardized and is controversial [32]. Most urologists define failure as the presence of >grade group (GG) 2 PCa and/or need for retreatment or radical treatment. However, it is important to distinguish between recurrences that occur in-field (within the ablated zone), indicating inadequacies in the technical intervention, such as insufficient margins or improper targeting, versus recurrences that occur out-of-field (outside of the ablated zone), indicating a failure in proper patient selection due to a missed lesion on imaging or biopsy or under-staging. Primary RT, serving as a similar alternative treatment option that has been around and studied much longer, has been associated with worse outcomes due to significant fibrosis that interferes with sRP, making it more difficult to perform, and with associated worsened postoperative and functional outcomes compared to pRP [33]. Proponents of FT have been hopeful that it would eliminate these concerns, lessening the morbidity and negative impact on quality of life, as it causes limited adverse effects on surrounding tissues. However, data is scarce, with limited knowledge on the outcomes and how it impacts subsequent sRP treatment in cases of cancer recurrence. As a result, it was the goal of our review to gather all the available evidence to be able to comment on the feasibility, safety, and effectiveness of sRP after prior failed FT. While there is no standardized definition for FT failure, the majority of patients in the selected studies underwent sRP if they had biopsy-proven PCa GG2 or higher, regardless of whether the lesions were located in-field or out-of-field. A rise in PSA or lesion changes on MRI raised suspicion for possible recurrence, and thus patients were subjected to biopsy to confirm. A few studies [18,20,21,23] included patients who additionally had a repeat ablation prior to sRP. We make a point to acknowledge a previous systematic review that was completed by Marra et al. in 2019, consisting of four retrospective studies including 67 men [34]. Our systematic review builds upon their prior work, with our study including 12 total studies, including 482 men that met inclusion criteria, describing postoperative, oncologic, and functional outcomes in patients completing sRP-FT. It is important to mention that sRP-FT is not the only available treatment for FT recurrence. Other possible treatments include sFT and sRT. However, we chose to focus on sRP-FT in this article. The majority of articles published for sFT are only on pRT failure patients, and those for sRT are on patients who received whole-gland FT only, not true “focal” ablations. Thus, comparisons cannot be accurately made.

### 3.8. Comparison with Other Studies

***Toxicity.*** We have reviewed the literature to see how our findings on sRP-FT compare to the available data on sRP-RT and pRP. Our compiled data across the 12 studies showed a total complication rate near 15% [0.09, 0.24] across 482 patients, with 4.6% accounting for major (CG > 3) complications. While Chade et al. [30] did not report on an overall complication rate in their systematic review of sRP after failed RT, they reported that the majority of complications were managed conservatively (CG 0-2), varying from 67 to 91%, similar to our findings for sRP after FT. They further reported that major complications (>CG3) varied from 0 to 25% in open sRP series, 0 to 11% in laparoscopic series, and 9 to 33% in robotic sRP series. Cohort studies in the literature reported total complication rates ranging from 13 to 27.5% for patients who received sRP-RT [35,36,37]. A systematic review and meta-analysis published by Valle et al. on salvage therapies after RT for PCa revealed a severe GU toxicity of 21% and severe GI toxicity of 1.9% in patients who underwent sRP-RT [38]. In comparison, our study revealed severe GU and GI toxicities in 3.6% and 0% of patients, respectively. For patients who received pRP, Tewari et al. reported an average complication rate of 12.3% in their systematic review, ranging from 7.8 to 17.9%. Interestingly, the rates were found to be dependent on the type of RP performed (open vs. laparoscopic vs. robotic), with open showing the worst (17.9%) rates and robotic (7.8%) showing the best rates [39]. Finally, Murphy et al. published the largest study of pRP patients, reporting an overall complication rate of 15.75%, with 5.25% being major complications (CG > 3), in 400 cases [40]. While it appears that complication rates can significantly vary between studies and with the type of surgery performed, we conclude that the complications are similar to those in patients who received sRP-RT or pRP. This suggests sRP-FT to be a safe treatment strategy, with no major concern or limitation that prior FT treatment impacts future sRP in case of recurrence. A possible explanation for the wide range of complication rates observed across studies may be related to surgeon experience and surgical technique (open vs. lap vs. robotic). Stephenson et al.’s work on open sRP-RT in and after 1993 found it to significantly reduce surgical morbidity, with fewer complications in later series, suggesting fewer complications with increasing center and surgeon experience [37]. Thus, experienced surgeons performing robotic sRP will likely lead to the least patient harm. It is advisable to inform patients undergoing focal therapy that it is not always completely effective and that they might require additional treatment, which may result in additional costs. We expect that there may be a change regarding the coverage of this treatment by insurance in the future. Previously, insurance did not cover it due to its experimental nature. However, as it is now widely used, we anticipate that insurance companies may begin to cover it, which would reduce the financial burden on patients.

***Oncological.*** Our meta-analysis demonstrated a PSM rate of 27% [0.19, 0.37] and a BCR rate of 23% [0.17,0.30] in patients undergoing sRP-FT. Thus, the overall BCR-free probability was 77% at a median follow-up time of 27 months. For patients with sRP-RT, Chade et al.’s systematic review reported a BCR-free probability of 47 to 82% across all selected studies at 5 years [30]. Their observed PSM rates varied from 43 to 70% in earlier series, with 0 to 36% in later series. Other studies describe a PSM between 10 and 19.2% [35,37,40]. Tewari et al.’s systematic review on pRP revealed an overall PSM average rate of 20.3% [39]. Murphy et al. published the largest study on pRP outcomes, with a PSM of 19.2%, and a BCR rate of 13.4% [40]. Taken together, the present data suggests sRP-FT to have slightly poorer oncologic control compared to pRP. While surgical difficulty after FT could have contributed to higher PSMs, it’s certainly possible that the patients who failed FT were in a poor prognostic category due to more severe intrinsic biology, and therefore fared worse after sRP. It is important to note that direct comparison is difficult considering disparate patient populations and eras. The nerve-sparing technique involves dissecting closer to the prostate gland to protect the neurovascular tissue, but doing so may also increase the chances of cutting into the tumor, which can result in a higher likelihood of PSM and cancer recurrence. Nguyen et al. conducted a study on 12 different research studies to evaluate the risk of PSM associated with both nerve-sparing and non-nerve-sparing approaches. Their findings indicated that for patients with pT2 disease, there was no significant difference in the risk of PSM between the two groups. However, for patients with pT3 disease, the nerve-sparing approach was associated with a significantly lower rate of PSM. Despite this, patients who underwent nerve-sparing surgery had a lower risk of experiencing urinary incontinence and erectile dysfunction compared to those who underwent non-nerve-sparing surgery [41].

***Functional.*** Our pooled data revealed a moderate detriment in urinary function, as 67% [0.53, 0.78] had complete (pad-free) urinary continence after sRP-FT treatment. The continence rates for sRP-RT, as seen in Chade et al.’s systematic review, ranged from 21 to 90% [30]. Rates were found to vary between 21 and 90%, 67 and 78%, and 33 and 80% in open, laparoscopic, and robotic, respectively. Similarly, another systematic review on sRP-RT by Parekh and colleagues found a 50% incontinence rate in 1329 patients across 24 series [42]. Other studies on sRP-RT showed 50–68% of patients maintained urinary pad-free continence by 12 months [35,36,37]. Murphy et al. showed a continence rate of 89.6% in pRP patients [40]. Taken together, our data suggests sRP-FT will result in moderate urinary dysfunction, performing better than sRP-RT but worse than pRP treatment. While unsure as to how sRP-FT causes a worsening of continence compared to pRP, it is important to consider the location of lesion targeting as a possible source for this difference. Particularly, Cathcart and colleagues [26] mention in their discussion that they attribute their better continence outcomes to the fact that no FT was performed around the apical portion, which ultimately led to sparing of the urethral sphincter. Additionally, it is important to note that our definition of continence was more conservative, defined as pad-free, whereas other studies may define continence as one pad/day, which may explain why our data had lower rates of continence compared to other studies. 

We observed poor overall potency across series, with an estimated 37% [0.18, 0.62] of patients achieving erections with or without medical treatment sufficient for penetrative sex. Chade and colleagues reported a significant detriment to EF, with 0 to 20% of patients maintaining EF after sRP-RT despite prepotency rates of 9 to 90% [30]. Similarly, other studies from the literature revealed EF recovery rates of 16 to 28% for sRP-RT despite adequate function prior to salvage treatment and nerve-sparing attempts [37,43,44]. Murphy et al. [40] reported a 62% recovery in potency in patients who were previously potent and underwent pRP. Comparing pRP vs. sRP-FT vs. sRP-RT, we conclude from the data that salvage treatment is associated with poor EF, regardless of prior treatment received. However, it is evident that pRP allows for better preservation of potency. We have one important consideration regarding our potency findings. Two studies [21,23] had 0% potency rates, which may have largely impacted the overall rate. The first study, by Thompson et al. [21], mentioned that 73% of patients were potent post-FT; however, none recovered function after sRP-FT. They mentioned that they were unable to complete nerve sparing in 60.8% of patients due to HIFU-induced fibrosis but did not provide further explanations as to why this resulted in poor EF outcomes. The study by Nunes-Silva [23] described significant impairment in EF, with an International Index of Erectile Function (IIEF)-5 score average of 22, decreasing to 3. They wrote that the likely effect was related to the energy directly applied to the prostate or to the high number of prostate biopsies during follow-up before salvage. 

FT also utilizes a spectrum of tissue-sparing templates, with no study commenting on the full details of locations where lesions were targeted—critical given the possibility that varying outcome differences are related to those who had lesions treated at the posterolateral zone adjacent to the neurovascular structures but could have otherwise had better outcomes had the lesions been treated elsewhere. Potency definitions and validated assessment tool use vary greatly across studies, also contributing to the difficulty in interpreting and comparing results. IIEF-5/SHIM is not used by all authors, and a certain score does not indicate full success in terms of potency. Those that did utilize such questionnaires pooled all the individuals’ data together into a mean score, which does not provide information as to whether each individual patient had erectile dysfunction or not. It is already known that sRP has a risk of affecting sexual function, but we believe that knowing the quantity of individuals affected provides more valuable information when counseling patients. To solve this issue, we encourage authors to utilize the standardized questionnaires for reporting functional outcomes in future articles. Furthermore, many studies focused on potency, which does not indicate good sexual quality of life/enjoyment. Other factors that are important for satisfaction but are never investigated are orgasm, ejaculatory function, sexual desire, and other masculinity/virility issues. 

While our data provides valuable information on the oncological and functional outcomes of sRP-FT, the study has some important limitations. The quality of current evidence is affected by the lack of well-conducted, randomized comparative studies, small study populations, and significant heterogeneity in terms of study design, study population, and assessment of primary outcomes. It is clear from our systematic review that further investigation is required, preferably through high-quality RCTs with more standardized/uniform comparisons (such as using only one FT modality and one RP technique) and longer-term follow-up to better evaluate and confirm the safety profile of sRP-FT. Additionally, this study identified HIFU as the most commonly used treatment modality, with data being more limited on other FT types; thus, future research will be needed to further investigate the effects of these other FT modalities to ascertain their relative safety and effectiveness. This will be critical, as it will ultimately determine treatment modality choice after FT failure for future PCa patients. 

## 4. Conclusions

FT as a treatment for localized PCa is continuing to be implemented in urologic practice, given the functional and quality of life benefits it has over the standard whole-gland treatment. However, with the increased risk of recurrence observed in FT patients, a small proportion will end up requiring sRP. Our systematic review provides insight that sRP-FT appears to be a reasonable treatment option for PCa recurrence, with acceptable complication rates and oncologic outcomes but significant morbidity for urinary function and EF compared to pRP. Future well-crafted studies will be required to fully assess the safety profile of such a treatment regimen.

## Figures and Tables

**Figure 1 cancers-15-02727-f001:**
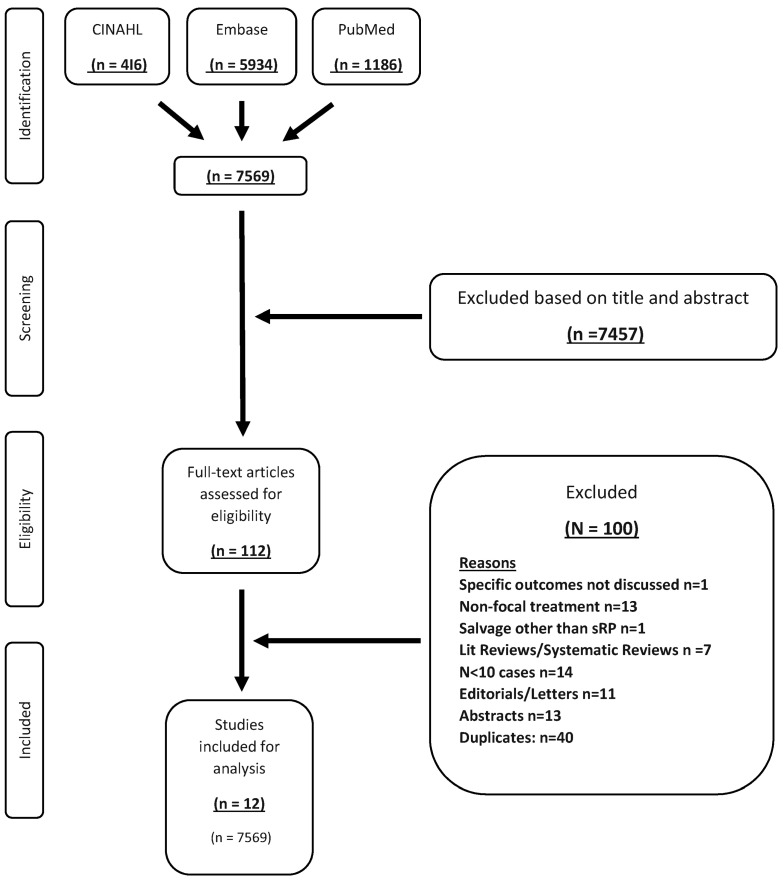
Preferred Reporting Items of Systematic Review and Meta-Analysis (PRISMA) flowchart detailing literature search and selection strategy for this study. sRP = salvage radical prostatectomy.

**Figure 2 cancers-15-02727-f002:**
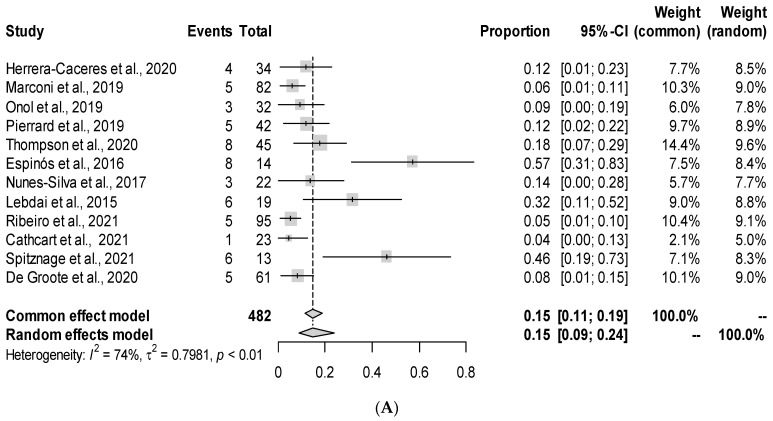
Forest plots showing individual study values of (**A**) the complication rate, (**B**) biochemical recurrence rate, (**C**) positive surgical margin rate, (**D**) continence rate, and (**E**) potency rate after salvage prostatectomy in patients with prior failed FT. CI = confidence interval [17,18,19,20,21,22,23,24,25,26,27,28].

**Table 1 cancers-15-02727-t001:** Risk-of-bias assessment using the Risk of Bias in Non-randomized Studies of Interventions (ROBINS-I) tool.

Author	Confounding Bias	Selection Bias	Measurement Bias	Deviations from Intended Intervention Bias	Missing Data Bias	Outcomes Measurement Bias	Selection of Reported Bias	Overall
Herrera-Caceres (2017) [17]	low risk	moderate risk	low risk	low risk	low risk	low risk	low risk	low risk
Marconi (2019) [18]	low risk	moderate risk	low risk	low risk	moderate risk	low risk	low risk	moderate risk
Onol (2020) [19]	serious risk	moderate risk	low risk	low risk	moderate risk	low risk	low risk	serious risk
Pierrard (2019) [20]	moderate risk	moderate risk	low risk	low risk	low risk	low risk	low risk	moderate risk
Thompson (2020) [21]	moderate risk	moderate risk	low risk	low risk	moderate risk	low risk	low risk	moderate risk
Linares Espinós (2016) [22]	moderate risk	moderate risk	low risk	low risk	low risk	low risk	low risk	moderate risk
Nunes-Silva (2017) [23]	low risk	moderate risk	low risk	low risk	low risk	low risk	low risk	moderate risk
Lebdai (2015) [24]	moderate risk	moderate risk	low risk	low risk	low risk	low risk	low risk	moderate risk
Ribeiro (2021) [25]	moderate risk	moderate risk	low risk	low risk	low risk	low risk	low risk	moderate risk
Cathcart (2021) [26]	moderate risk	moderate risk	low risk	low risk	low risk	low risk	low risk	moderate risk
Spitznagel (2021) [27]	moderate risk	moderate risk	low risk	low risk	low risk	low risk	low risk	moderate risk
De Groote (2020) [28]	moderate risk	moderate risk	low risk	low risk	low risk	low risk	low risk	moderate risk

**Table 2 cancers-15-02727-t002:** Compiled results across all the studies. Postoperative complications, oncologic outcomes, and functional outcomes.

First Author	Study Type	Yr	No of Pts Who Received FT	Age (Yr)	Avg Follow- Up (mo)	Time from Tx to sRP (mo)	% Total Complications	% Total >CG3	% PSM	% BCR	% Continent	% Potent
Herrera-Caceres [17]	Retrospective noncomparative	2020	34	61	52	11	12%	12%	38%	21%	91%	53%
Marconi [18]	Retrospective noncomparative	2019	82	65	-	27	6%	1%	13%	41%	83%	21%
Onol * [19]	Retrospective comparative	2020	32	66	29	61	9%	3%	44%	19%	78%	28%
Pierrard [20]	Retrospective noncomparative	2019	42	65	23	17	12%	2%	31%	10%	64%	75%
Thompson [21]	Retrospective noncomparative	2020	45	63	18	30	18%	2%	44%	24%	67%	0%
Linares Espinós * [22]	Retrospective comparative	2016	14	65	62	24	57%	21%	7%	29%	56%	60%
Nunes-Silva ** [23]	Retrospective comparative	2017	22	63	-	24	14%	14%	5%	32%	32%	0%
Lebdai [24]	Retrospective noncomparative	2015	19	64	10	17	32%	11%	47%	0%	68%	91%
Ribeiro * [25]	Retrospective comparative	2021	95	65	30	36	5%	1%	13%	32%	83%	20%
Cathcart [26]	Retrospective noncomparative	2021	23	63	-	25	4%	0%	35%	17%	83%	64%
Spitznagel *** [27]	Retrospective comparative	2021	13	61	-	15	46%	31%	8%	0%	23%	89%
De Groote * [28]	Retrospective comparative	2020	61	67	25	**-**	8%	2%	38%	18%	39%	5%

* FT vs. RT; ** FT vs. pRP; *** sRP vs. pRP. BCR = biochemical recurrence, CG = Clavien grade, FT = focal therapy, PSM = positive surgical margin, pRP = primary radical prostatectomy, sRP = salvage radical prostatectomy, Tx = treatment.

**Table 3 cancers-15-02727-t003:** Baseline patient characteristics and salvage prostatectomy outcomes.

	Value
Median age at time of sRP (yrs)	65 (range 61–67)
Time from primary treatment to sRP (mo)	24 (range 11–61)
Median follow up (mo)	27 (range 10–62)
Type of FT prior to sRP (*n* = 482)	
HIFU	295 (61.2%)
Cryotherapy	79 (16.4%)
VTP	69 (14.3%)
Irreversible electroporation	17 (3.5%)
Laser	15 (3.1%)
PRX302 Topsalysin	4 (0.8%)
Brachytherapy	3 (0.6%)
Repeat ablation (*n* = 248)	
YES	37 (14.9%)
NO	211 (85.1%)
SRP type (*n* = 468)	
Robotic	399 (85.3%)
Open	60 (12.8%)
Laparoscopic	9 (1.9%)
Nerve-sparing (*n* = 468)	
Unilateral	145 (30.9%)
Bilateral	115 (24.6%)
None	208 (44.5%)
Pre-Preoperative sRP Biopsy (*n* = 482) *	
Grade Group I (3 + 3)	84 (17.4%)
Grade Group II (3 + 4)	216 (44.8%)
Grade Group III (4 + 3)	119 (24.7%)
Grade Group IV (4 + 4)	21 (4.4%)
Grade Group V (4 + 5, 5 + 4)	17 (3.5%)
Missing	9 (1.9%)
Postoperative outcomes	
Total number of complications (*n* = 482)	59 (12.2%)
Clavien grade I	24 (40.7%)
Clavien grade II	13 (22.0%)
Clavien grade IIIa	7 (11.9%)
Clavien grade IIIb	14 (23.7%)
Clavien grade IV	1 (1.7%)
Clavien grade V	0 (0%)
Major complications (CG >3) (*n* = 482)	18 (30.5%)
Oncologic outcomes (*n* = 482)	
Positive surgical margin	126 (26.1%)
Biochemical recurrence	118 (24.5%)
Postoperative sRP Biopsy (*n* = 482) *	
Grade Group I (3 + 3)	40 (8.3%)
Grade Group II (3 + 4)	215 (44.6%)
Grade Group III (4 + 3)	134 (27.8%)
Grade Group IV (4 + 4)	15 (3.1%)
Grade Group V (4 + 5, 5 + 4)	27 (5.6%)
Missing	1 (0.2%)
Functional outcomes	
Continence (12 mo) (*n* = 467)	319 (68.3%)
Potency (12 mo) (*n* = 351)	107 (30.5%)

CG = Clavien grade, FT = focal therapy, HIFU = high-intensity focused ultrasound, sRP = salvage radical prostatectomy, VTP = vascular targeted photodynamic therapy. * One study, Onol et al., combined GG2-3 together (*n* = 20) and GG4-5 (*n* = 6) together for preop sRP biopsy. Similarly, they combined GG2-3 together (*n* = 23) and GG4-5 (*n* = 6) together for postop sRP specimen pathology.

## Data Availability

Compiled data supporting results can be obtained from authors upon request.

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
