# Peer review of "Salvage Radical Prostatectomy after Primary Focal Ablative Therapy: A Systematic Review and Meta-Analysis"

_cancers, 2023, doi:10.3390/cancers15102727_

Round 1
Reviewer 1 Report
The authors conduct a systematic review of salvage radical prostatectomy after different focal therapy for prostate cancer. The intent is interesting even if the large number of data complicates the linearity among the objective, the results and the conclusion. The authors evaluate a number of focal therapies, but HIFU almost always emerges in the manuscript. It would be interesting to understand the data that emerge in a direct comparison between the various focal techniques and the correlation with post-surgical side effects. I don't have Tables or Graphs and I think my confusion about manuscript stems from this problem. I ask to review the manuscript with the suggestions I have added and with the graphs and tables. Thank you
Reviewer 2 Report
The authors present an interesting review about the safety and results of sRP following PCa recurrence after FT. Of course, since the FT is not many years done, the studies cannot be of best quality.
The citations are not referred in the same manner through the text.
Reviewer 3 Report
Dear Authors,
I would like to congratulate you on very interesting work. It allows to systematize current knowledge on Focal Treatment in prostate cancer. It is a method, that can have well established position in the treatment of this common disease. I recommend the manuscript to the further editing process, taking a few remarks into consideration. I would like to ask for:
- an explanation why the terms ‘historically’ and ‘favored’ are used in the lines 54 and 55. According to the current EAU Guidelines, radical prostatectomy and radiotherapy are still recommended for localized prostate cancer. These guidelines are strongly supported by research, not emotion. The use of the words "historically" and "favored" by the authors is biased and suggests that their analysis of the literature is also partial. It is also worth pointing out that the EAU Guidelines in intermediate-risk prostate cancer recommend: “Only offer whole-gland ablative therapy (such as cryotherapy, high-intensity focused ultrasound, etc.) or focal ablative therapy within clinical trials or registries /strong/”. Thus, the viewpoint of the urological bodies should also be discussed.
- in “discussion” part, I expect adding a paragraph about the financial aspect related to the necessity of treating second-line patients after FT failure,
- discussion about higher rate of positive margins after sRP in relation to the technique of nerve-sparing surgery - over 50% of patients operated on with this method. Is it safe for the patient? Should the qualification be similar to that of patients scheduled for primary prostatectomy?
- unifying page numbers in References section, for instance position no. 40 and 43 are different,
- position no. 45 in References section contains mistakes starting from place- year of publication
Good luck
Reviewer 4 Report
Authors should be congratulated for their thoroughly systematic review and metanalysis. Overall methodology is correct and there are no major concerns. Focal therapy is an increasingly adopted therapy to treat prostate cancer. Authors focus on the surgical salvage treatment showing its feasibility in terms of cancer control. Indeed, functional results are suboptimal. The paper could help counsel patients.
Round 2
Reviewer 1 Report
No other changes are required.
Author Response
Dear Reviewer,
Thank you for taking the time to review our paper. We appreciate your feedback and are delighted to hear that you found our research to be well-conducted and well-presented. Your positive evaluation is a great encouragement for us. We would like to express our gratitude for your valuable input, and we will take your comments into account in our future work.
Once again, we appreciate your time and effort in reviewing our paper and look forward to the opportunity to share more of our research with you in the future.
Best regards,
Fernando